# Improved Memory and Lower Stress Levels in Male Mice Co-Housed with Ovariectomized Female Mice

**DOI:** 10.3390/ani14101503

**Published:** 2024-05-18

**Authors:** Layung Sekar Sih Wikanthi, Johan Forsström, Birgit Ewaldsson, Vilborg Palsdottir, Therése Admyre

**Affiliations:** 1Department of Animal Science and Technology, Clinical Pharmacology&Safety Science, R&D, AstraZeneca, 43183 Gothenburg, Sweden; layung.wikanthi1@astrazeneca.com (L.S.S.W.); birgit.ewaldsson@astrazeneca.com (B.E.); 2Department of Translational Genomics, Discovery Sciences, R&D, AstraZeneca, 43183 Gothenburg, Sweden; johan.forsstrom@astrazeneca.com (J.F.); vilborg.palsdottir@astrazeneca.com (V.P.)

**Keywords:** male mice aggression, housing condition, group-housed, paired with ovariectomized female, anxiety-like behavior, stress level, corticosterone

## Abstract

**Simple Summary:**

Fighting is the most common problem in group-housed male mice. Fighting leads to stress, wounds, and sometimes result in death, which is against animal welfare and contradicts the 3Rs principle. In this study, we aimed to evaluate behavior and stress levels in group-housed males and males pair-housed with ovariectomized female mice. Our study showed that whilst some group-housed males do fight, no male pair-housed with ovariectomized females does. In addition, pair-housed males had a better memory, and less anxiety-like behavior. Pair-housed male mice had a larger reduction in corticosterone levels during the study, indicating lower stress levels. These findings suggest that pair-housing male mice with ovariectomized females could refine the housing conditions for laboratory male mice.

**Abstract:**

Aggressiveness, expressed by fighting, is a frequent problem in group-housed laboratory male mice and results in increased stress, injury, and death. One way to prevent fighting is by pairing the male mice with ovariectomized female mice to provide a compatible companion. However, the effect of these housing conditions remains unclear. Therefore, we aimed to evaluate behavior and stress levels in two different housing conditions, pair-housed with an ovariectomized female and group-housed with other males. Behavioral tests were performed to assess stress and anxiety-like behavior. Moreover, the corticosterone levels in plasma were measured by ELISA. Based on home cage behavior assessment, pair-housed male mice showed no signs of fighting, not even after isolation and regrouping. Our results also showed that the pair-housed males had a better memory and demonstrated less anxiety-like behavior. Subsequently, the pair-housed male mice had a larger reduction in corticosterone levels compared to group-housed males. Overall, pair-housing reduced anxiety-like behavior and stress levels in male mice compared to standard group-housing.

## 1. Introduction

Animal models are commonly used for scientific purposes, particularly in biomedical research [1]. Using animal models is beneficial to study the mechanisms underlying human diseases and predict the effect of a candidate drug before applying it to humans [2]. Animal studies provide knowledge that could lead to new findings on diagnosis and treatment. Mice are extensively used in experimental studies since their genetics, anatomy, and physiology are similar to humans [3]. Mice can also be genetically modified to mimic human diseases, making it possible to model several diseases such as Alzheimer’s disease, Huntington’s disease, different types of cancers, and many other diseases [3,4,5]. Furthermore, in some disease models, depending on the research approach, male mice are more regularly used compared to female mice. One reason for this is that male mice have more stable hormone levels since they are not influenced by the estrous cycle [6,7]. However, the usage of male mice is sometimes problematic. Aggressiveness is one of the most serious problems that researchers face when group-housing male mice. Male mice form a complex social structure within a group [8,9]. However, the presence of intruders in the group can influence the stability of dominance hierarchy, which leads to further aggression. A dominance hierarchy determines the access to food, females, and territory. The dominant male produces a high number of offspring during breeding, while the subordinates can hardly mate and breed [10,11]. However, due to social conflicts, the dominant male tends to attack the subordinate males, resulting in fighting [12], which is one of the most common forms of aggressiveness. It has been reported that aggression is the most frequent problem during husbandry [13], which conflicts with animal welfare. Aggression potentially drives the mice into stress, pain, injury, and even, in the worst case, death. Besides animal welfare concerns, aggression induces alteration in physiological, behavioral, and pathological parameters [14,15], resulting in data invalidity and increased statistical variation [16,17,18]. Therefore, researchers should apply the principle of 3R’s (Replacement, Reduction, and Refinement) to ensure animal welfare and production of reliable data.

The principles of the 3Rs were developed over 50 years ago by W.M.S Russell and R.L. Burch as a framework for humane animal research. The first principle, replacement, means avoiding animal usage and using alternative methods such as in vitro or computer-based models when possible [16]. If the usage of animals cannot be avoided, the researcher should consider a reduction of the number of animals. The third principle is refinement, which means reducing the level of distress and preventing unnecessary pain during a study. Refinement in the form of providing proper husbandry, optimal housing conditions, and compatible cage mates could address the problem related to aggressiveness in the home cage.

Providing proper husbandry in the cage prevents instability. Enrichment such as nesting material has been reported to decrease intermale aggression [13,19]. Additionally, housing conditions that include a group size to promote social hierarchical stability would help, but there is no clear evidence for an optimal group size of male mice. Some studies reported that group-housing three to four male mice in a standard cage would enhance the social hierarchical stability that prevents aggression [9,20,21,22]. In addition, it has been reported that group-housing two intact male mice, as well as placing an intact male with a castrated male mouse in the same cage, resulted in fighting [23]. In addition, grouping males before they are sexually matured and grouping close relatives (siblings) could prevent severe fighting [24].

Another way to prevent severe injury due to aggression is to single-house male mice. Single-housing could be required if injury occurs due to social conflicts in the cage [13] and in certain experimental settings, for example, during an individual food intake study or after certain surgical procedures [25,26]. However, single-housing mice leads to stress-induced changes in their behavior and physiology [27,28]. Since mice are social animals, frequent periods of isolation can induce stress [29], and therefore, a housing condition that supports social interactions is preferable.

Considering the social need for a companion, an ovariectomized female mouse could be a better housing partner for male mice than other male mice. One study reported that ovariectomized females are the best companions for male mice to prevent fighting and any aggressive-like behavior [23]. In addition, using ovariectomized females instead of fertile females is also a way to prevent uncontrolled breeding in the animal unit. In several of our studies, we do not have ethical permission to single-house male mice, and we are, at the same time, experiencing problems with fighting in group-housed male mice. Our alternatives for housing are therefore to either try to keep the male mice group-housed (if they do not hurt each other) or pair-house them with an ovariectomized female. However, the optimal housing conditions and cage companions still remain unclear, and the behavior and stress levels in pair and group-housed male mice have not been investigated previously. In the present study, we aim to evaluate the effect of different cohousing conditions by performing behavioral tests and measuring stress hormone levels.

## 2. Materials and Methods

### 2.1. Animals, Housing, and Termination

All animal experimental protocols were approved by the Gothenburg Ethics Committee for Experimental Animals (license numbers: 3073-2020 and 3092-2020), compliant with EU directives on the protection of animals used for scientific purposes. Animals were held in an AAALAC (Association for Assessment and Accreditation of Laboratory Animal Care International) facility. The overall study design is shown in Figure 1. 

Thirty-six C57BL/6NCrl male mice were obtained from Charles River (Sulzfeld, Germany) at five weeks of age. The male mice were already housed in groups of four from weaning age and kept in these groups until the study started. The ovariectomized C57BL/6NCrl female mice (Charles River, L’Arbresele, France) were housed in groups of six upon arrival and kept in a room for females before pair-housing them with the males. Ovariectomy surgery was performed by Charles River (L’Arbresele, France). Before surgery started, the surgical area was shaved and disinfected with an aseptic solution. Eye gel was applied to protect the eyes of the mice. Analgesia (Buprenorphine at 0.1 mg/kg) was injected subcutaneously before surgery. The mice were anaesthetized with isoflurane. Ovariectomy surgery was performed by making a dorsal incision in the skin. A blunt puncture through the abdominal wall was performed on each side and the ovaries were removed. The fallopian tube was cauterized subsequently, and the skin incision was closed with wound clips. After surgery, clinical examination was performed at the same occasion as injection of analgesia. Buprenorphine was injected 4 h after surgery. In addition, NSAIDs (Carprofen at 5 mg/kg) were injected subcutaneously 24 h and 48 h after surgery. A heating pad was utilized during the whole recovery time. The ovariectomized female mice were recovered after 3 to 5 days. 

After one week of acclimatization, the male mice (six weeks of age) were divided into two groups consisting of twenty-four male mice that were kept group-housed in groups of four and twelve male mice that were paired with ovariectomized C57BL/6NCrl female mice (Charles River, L’Arbresele, France). All mice were kept in macrolon 3 cages (800 cm^2^ and 18.5 cm high) equipped with aspen chips (Tapvei, Harjumaa, Estonia) and nesting material including shredded paper, gnaw sticks, and cotton rolls. The use of extra environmental enrichment such as a cardboard house or tunnel was avoided in the group-housed mice to prevent territorial marking but included in the cages with pair-housed mice (male and ovariectomized female).

The animals were maintained under a constant temperature (21 °C) with a 12:12 h light–dark cycle (lightening: 05.30–06.00 am, darkening: 05.30–06.00 pm) and a humidity of 40–50%. They had free access to standard R70 chow diet (Lantmännen Lantbruk, Kimstad, Sweden) and water ad libitum. Cage changes were conducted every week or if needed when bedding material was wet. The mice were checked daily, to monitor any signs of fighting, and weighed weekly. At the end of the experiment, mice were anesthetized with 5% isoflurane, shaved to check for fighting scars, and terminated by decapitation.

### 2.2. Home Cage Behavior

Home cage behavior was assessed by visual observation focused on identifying any signs of fighting including attacking, chasing, biting, quarrelling, and threatening posture. The first home cage observation period was when male mice were first placed together with their ovariectomized female. Mice were observed for the first twenty minutes, 1 h after pairing, and at 2 pm in the afternoon of the day of pairing. Further observations were performed every morning (8 am) and afternoon (2 pm) during the 3 following days post pairing, and vaginal plug check was performed every morning at the ovariectomized females during those 3 days. Meanwhile, the group-housed male mice were monitored daily throughout the entire study to assure that fighting had not occurred.

Home cage behavior monitoring was again performed when the male mice were returned after a period (3 days) of being single-housed. For both pair- and group-housed mice, home cage behavior was monitored during the first hour after regrouping and at two more timepoints that same day, and then every morning for 3 consecutive days. The mice were checked for biting marks by stroking the fur. Plug checks were performed in ovariectomized females every morning for 3 consecutive days after regrouping them with the males.

### 2.3. Passive Avoidance Test (PAT)

The passive avoidance test (PAT) aims to evaluate fear-conditioned memory by using an unpleasant stimulus [30,31]. It was performed in a shuttle box system (Accuscan Instruments Inc., Columbus, OH, USA) which contains two compartments separated by a sliding door. One of the compartments has transparent walls (the light compartment) and the other compartment has opaque walls (the dark compartment). PAT was performed on two consecutive days. On the first day, the mouse was placed into the light compartment, and after 30 s, the sliding door between the light and the dark compartment was automatically opened. Mice tend to enter the dark compartment as they naturally prefer dark places [32]. Once the mouse entered the dark compartment, the sliding door was closed, and the mouse was given a mild electric shock (0.3 mA) through the grid. On the next day, the mouse was again placed into the light compartment and the latency to enter the dark compartment was recorded. A longer latency to enter the dark compartment on the second day was interpreted as better memory of the unpleasant stimuli.

### 2.4. Zero Maze Test

The zero maze test was performed to study anxiety-related behavior in the mice [33,34]. The zero maze (Accuscan Instruments Inc., Columbus, OH, USA) is a circular black acrylic platform which is 5 cm wide, has an inner diameter of 40 cm, and is elevated 75 cm from the floor. The maze is equally divided into four areas consisting of two opposite sides which are covered by 30 cm acrylic transparent walls (closed arms) and two other arms that are open. The maze is equipped with photocell transceivers which monitor the mouse’s activity when it goes in and out from the closed arms. During the experiment, the mouse was placed on one of the closed arms, and the test was performed for 5 min on two consecutive days, on the first day for measurements in a novel environment and on the second day in a familiar environment. The activity in the closed and open arms, the time spent in both arms, and the latency to enter the open arm for the first time were recorded. Mice who spent more time in the closed arms were interpreted as more anxious than mice who spent less time in the closed arms. 

### 2.5. Corticosterone Measurement

Blood was collected for corticosterone measurement in the beginning of study (when mice were 6 weeks of age) and before terminating the mice. Blood sampling was performed between 09.00 am and 11.00 am both for baseline and terminal sampling. Mice were placed in a restrainer (Agnthos AB, Lidingö, Sweden), the distal part of the tail vein was briefly cut using a sterile scalpel blade, and 20 µL blood was collected in EDTA tubes (Satstedt, Nümbrecht, Germany). After the tail cut, the bleeding was stopped by applying gentle pressure with a sterile pad.

All blood samples were centrifuged at 5000× *g* for 10 min in 4 °C to separate plasma. The plasma samples were then stored in −20 °C until analysis. Plasma was analyzed using corticosterone Enzymed Linked Immunosorbent Assay (ELISA) kit (Arbor Assays, Ann Arbor, MI, USA). The procedure was performed according to the protocol provided by the manufacturer.

### 2.6. Statistical Analysis

Normal distribution analysis was performed for all data using the Shapiro–Wilk test. Unpaired t-test was used to compare differences between the two groups (pair-housed males and group-housed males) when data were normally distributed. Mann–Whitney U test was used for the data that were not normally distributed. All data were calculated in GraphPad prism, and a *p* < 0.05 was considered to be statistically significant.

## 3. Results

### 3.1. No Fighting Was Found in Pair-Housed Males after 3 Days of Separation

No fighting was observed between male mice and ovariectomized female mice when they were first put together in the same cage. During the first hour after cohousing, the male mice tried to mate and followed the ovariectomized female mice (courtship). Some of the pairs were mounting and sniffing each other. When observing them later the same day and on the following 3 days, all pair-housed mice stayed together in a common nest and there were no signs of fights or bite marks. Likewise, while regrouping after 3 days of separation, the pair-housed mice showed no signs of fighting, no bite marks were found, and the pairs stayed together in the nest. In addition, the ovariectomized females had no vaginal plugs after pair-housing the second time.

The group-housed mice showed no signs of aggression or fighting before the 3-day separation. In one out of six cages, fighting started 5 min after regrouping the mice. To avoid further stress and physical wounds, the mice in this cage were immediately separated and single-housed. These four male mice were excluded from the following tests. Meanwhile, the mice in the other five cages did not show any signs of fighting after regrouping them with their former cage mates. The mice showed rearing, climbing, digging, anogenital recognition, and grooming as signs of exploring the cage mates again. During the afternoon, the first day and the 3 following days of observations, no fighting was found in the remaining group-housed cages, and the mice stayed together in a common nest.

### 3.2. Both Groups Showed an Increase in Body Weight over Time, But No Significant Difference Was Found between the Groups

Body weight was recorded once per week during the 7-week long study (Figure 2). The body weight increased over time in both groups. However, there was no significant difference in body weight between pair-housed and group-housed mice.

### 3.3. The Pair-Housed Males Displayed a Better Memory than the Group-Housed Males

All mice entered the dark compartment in the passive avoidance test on day 1 within the 5 min time frame (Figure 3a), and all mice could therefore be included in the memory assessment on day 2. The pair-housed mice had a significantly longer latency to enter the dark compartment compared to the group-housed mice on the second day (Figure 3b).

### 3.4. The Pair-Housed Mice Were Less Active in the Zero Maze

On the first test day of the zero maze, the pair-housed mice showed lower activity, while no differences were seen in latency to enter the open arm (Figure 4a). However, on the second day, pair-housed mice showed significantly longer latency to enter the open arm for the first time (Figure 4b). Our results also showed that pair-housed mice tended to stay longer in the open arms and for a shorter time in the closed arms compared to the group-housed mice on both days.

### 3.5. The Pair-Housed Mice Have Significantly Larger Reductions of Corticosterone Levels

Corticosterone levels were significantly different between the pair-housed and group-housed male mice. The reduction in corticosterone levels was significantly larger in the pair-housed males than in the group-housed males Figure 5).

## 4. Discussion

According to our understanding, different housing conditions affect behavioral and stress hormone levels in mice, but this has not been sufficiently studied and not reported in the literature previously. Previous studies had their focus on home cage behavior without evaluating performance in specific behavioral tests [23] and without investigating stress hormone levels during different housing conditions [28]. There are a limited number of studies comparing group-housed male mice and male mice pair-housed with ovariectomized female mice [23]. Therefore, we aimed to provide new findings by assessing specific behaviors and stress hormone levels in these two different housing conditions consisting of group-housed male mice and male mice pair-housed with ovariectomized female mice. We found that the pair-housed male mice demonstrated a better memory, showed larger decrease in corticosterone hormone levels in plasma from baseline to the end of the study, and had a tendency of decreased anxiety-like behavior compared to the group-housed male mice.

The home cage behavior assessment showed that male mice tried to follow and mate with the females during the first hour of pair-housing, which is a normal sign of courtship [35]. No signs of fighting were found in the pair-housed cages, and all pairs stayed together in a common nest, indicating that ovariectomized female mice seem to be compatible companions for the male mice. Notably, after being apart for 3 days and then regrouped again, there were no signs of aggression or fighting in the pair-housed mice. Additionally, vaginal plugs were absent in the ovariectomized female, meaning that mating did not occur. Ovariectomized females might contribute to reduce intermale aggression [36], and these results align with the previous study, indicating that they are good companions for male mice [23].

On the other hand, in the group-housed mice, fighting was found in one out of six cages after separation and regrouping. Many factors are involved in fighting, such as male pheromones enhancing inter-male aggression in the groups [37], (re)establishing of social hierarchy [11], or external factors of husbandry conditions. The remaining group-housed mice stayed together in the nest, indicating that they had formed compatible groups [38]. In addition, it seems that the way we housed the mice, by grouping them four per cage [9,20] and keeping the old unsoiled bedding material between cage changes [39], contributed to lower aggression. This finding pinpoints the importance of refined environmental conditions, which could prevent emotional instability in the animals [40]. In addition, the body weight monitoring (Figure 2) showed no differences between pair-housed and group-housed mice, indicating that no animals were severely stressed or in pain [41].

In the passive avoidance test, the pair-housed mice took a longer time to enter the dark compartment during the second day, indicating that the pair-housed mice had a better memory of the unpleasant shock in the dark compartment from the first day compared to the group-housed male mice (Figure 3b). Mice with a normal functioning memory stay longer in the light compartment since they remember the unpleasant shock, have learned to avoid the black compartment, and associates it with the unpleasant stimuli [32,42]. Group-housed male mice form a hierarchy within the group, which also affects the stress levels of the mice, depending on their ranking in the group. There is a clear link between social stress and memory [43], which means that the social stress caused by the group-housing condition can be the reason for the lower performance in the memory test compared to the males that were pair-housed with ovariectomized females.

In the zero maze test, the pair-housed mice showed significantly lower activity during the first test day compared to the group-housed mice, which could indicate less stress in a novel environment (Figure 4a). It seems that the pair-housed mice tend to stay longer in the open arm, during both test day 1 and 2, than the group-housed mice, indicating lower anxiety. In addition, a longer latency to enter the open arm on the second day, in a familiar environment, could be an indication of lower stress levels rather than indicating that the pair-housed mice are more anxious since the environment is familiar and there is no need to explore it to the same extent as in the first day when the environment is novel. Lastly, there was a significant difference in the change of corticosterone levels between the two experimental groups. The pair-housed group had significantly larger reduction of corticosterone level from baseline to the end of the study. The finding indicates that the pair-housed mice were less stressed than the group-housed mice. Together, the males pair-housed with ovariectomized females had lower stress levels and a tendency towards lower anxiety compared to group-housed male mice.

In addition, the ovariectomized females were observed to nest and engage with the fertile males. No plug was found in the females, which may suggest that the males did not try to mate with them. Overall, home cage behavior observation of the females may suggest that co-housing with fertile male mice had limited impact on their welfare. However, the females still underwent surgery that could had affected their welfare. In our research facility, ovariectomized females were not subjected to any other procedure and will be used throughout out their lifetime to reduce the number of animals needed. However, it remains important to balance animal welfare against the expected gains. We aim to further explore this balance in future studies.

Our study presents a novel cohousing alternative that prevents fighting in male mice. These findings could be used to improve the physical and psychological welfare of experimental animals. Co-housing male mice with ovariectomized female mice could also reduce data variability within a study by improving animal welfare. However, there is, of course, an ethical dilemma in using extra female mice that undergo a small surgical intervention to increase the animal welfare of the single-housed laboratory male mice. By using surplus females from our breeding unit and using them as companions in several studies throughout their life span, we could maximize the benefit of each animal. In future studies, we could broaden the understanding of the best housing conditions for both genders by investigating the stress levels in the female companion mice as well.

## 5. Conclusions

In this study, we demonstrated how the cohousing condition of C57BL/6NCrl male mice affected their behavior and stress hormone levels. Male mice pair-housed with ovariectomized female mice did not show any aggressiveness or fighting behavior. In addition, male mice from the pair-housed group had a better memory, larger reduction in stress hormone levels, and tended to be less anxious than the group-housed male mice. Subsequently, cohousing male mice with ovariectomized female mice could be an option to refine the environmental factors and reduce stress and anxiety in laboratory male mice.

## Figures and Tables

**Figure 1 animals-14-01503-f001:**
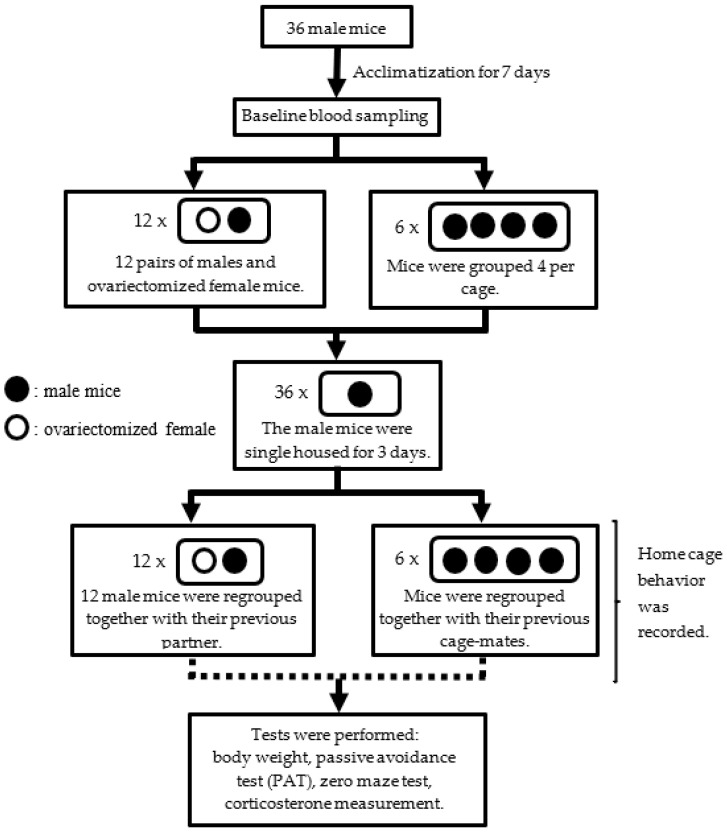
Experimental design. Male mice were divided into two groups consisting of pair-housed with ovariectomized female mice (*n* = 12) and group-housed with males (*n* = 24). At one time point, all mice were separated from their cage mates for 3 days and regrouped again. In the event of fighting in any cage, the mice were split up and excluded from following tests.

**Figure 2 animals-14-01503-f002:**
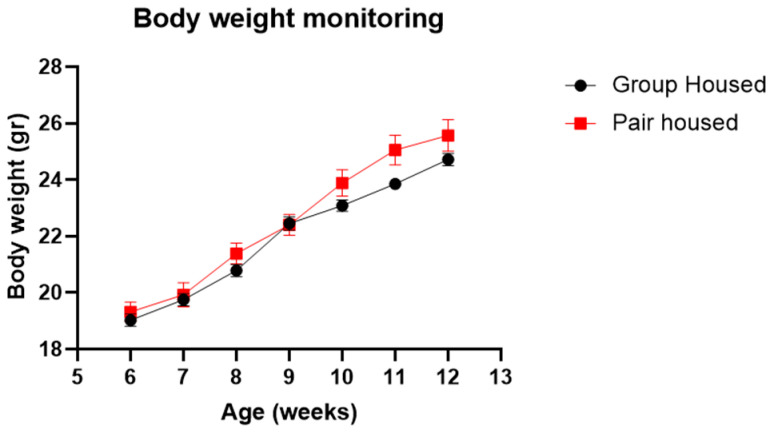
Body weight monitoring of group-housed and pair-housed mice (n_group-housed_: 20, n_pair-housed_: 12) between 6 and 12 weeks of age.

**Figure 3 animals-14-01503-f003:**
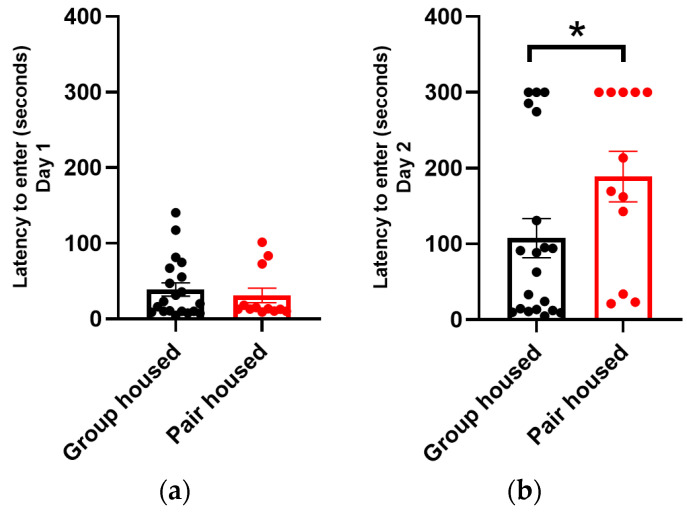
Latency to enter the dark compartment during passive avoidance test (PAT). Values are calculated as mean ± SEM (n_group-housed_: 20, n_pair-housed_: 12). (**a**) PAT Day 1. (**b**) PAT Day 2. * *p* value < 0.05.

**Figure 4 animals-14-01503-f004:**
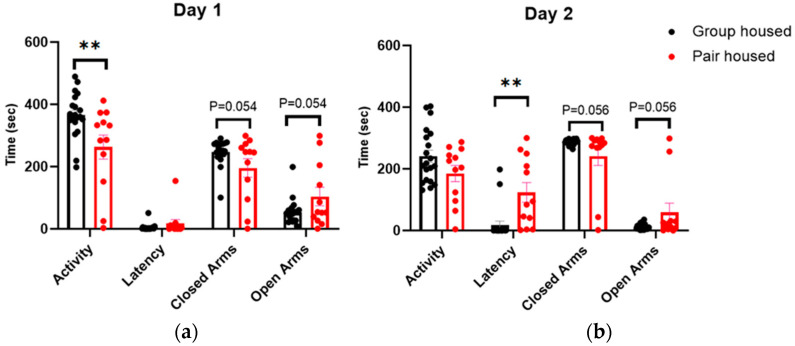
Activity, latency to enter the open arm, time spent in the closed arms and time spent in the open arms in the Zero maze test. Data expressed as mean ± SEM (n_group-housed_: 20, n_pair-housed_: 12). (**a**) Zero maze test in Day 1. (**b**) Zero maze test in Day 2. ** *p* value < 0.01.

**Figure 5 animals-14-01503-f005:**
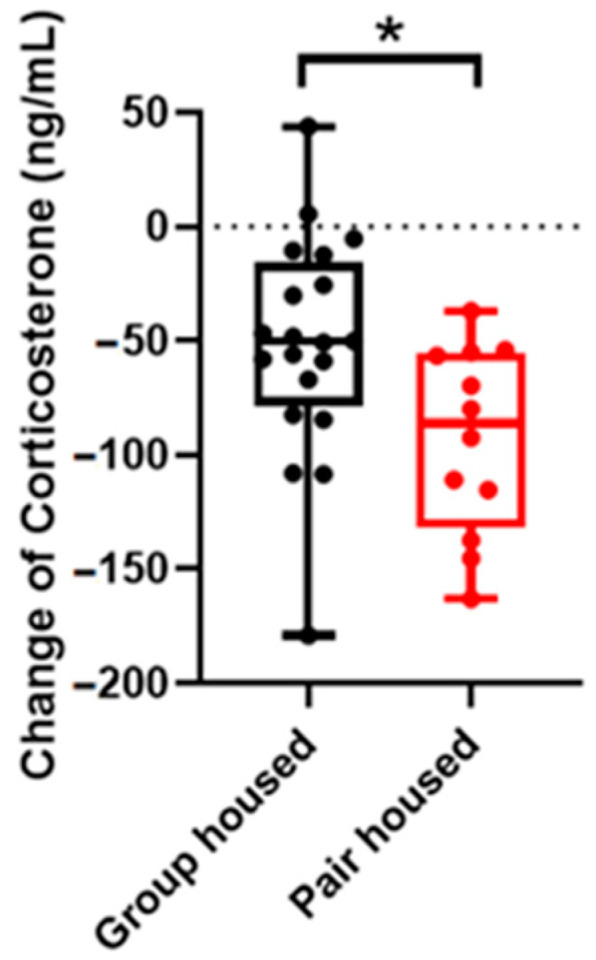
Changes of corticosterone levels in plasma from baseline to end of study (n_group-housed_: 20, n_pair-housed_: 12). Data expressed as median, minimum, and maximum points. * *p* value < 0.05.

## Data Availability

All data generated or analyzed during this study are included in this published article.

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
