# Peer review of "Improved Memory and Lower Stress Levels in Male Mice Co-Housed with Ovariectomized Female Mice"

_animals, 2024, doi:10.3390/ani14101503_

Round 1

Reviewer 1 Report

Comments and Suggestions for Authors

This manuscript overs an interesting topic. It is well organized and written. The figures are useful and appropriate. There is go use of the scientific literature, although I noticed that the journal name wasn't provided for citation #23 (Ewalksson et al. 2010). But my main concern is that I'm not really convinced that the results and stats justify maintaining 4 cages to maintain 4 males for research versus maintaining 4 males in a single cage, especially when the females placed with each male need to be ovary-vasectomized. It seems the costs would be much higher given very little improvement in animal condition and behavior. Perhaps this could be explained a little more in the conclusions section ?!? 

Author Response

This manuscript overs an interesting topic. It is well organized and written. The figures are useful and appropriate. There is go use of the scientific literature, although I noticed that the journal name wasn't provided for citation #23 (Ewalksson et al. 2010)

We apologize for missing the complete citation for Ewaldsson et al, 2016 and we have now added the journal name. The complete citation is found in line 507-508: “Ewaldsson B, Nunes F, Gaskill B, Ferm A, Stenberg A, Pettersson M, et al. Who is a compatible partner for a male mouse? Scandinavian Journal of Laboratory Animal Science. 2016;42

 But my main concern is that I'm not really convinced that the results and stats justify maintaining 4 cages to maintain 4 males for research versus maintaining 4 males in a single cage, especially when the females placed with each male need to be ovary-vasectomized. It seems the costs would be much higher given very little improvement in animal condition and behavior. Perhaps this could be explained a little more in the conclusions section ?!? 

We are aware of that the study design can be discussed, but for us it was important to compare the housing conditions that we are using in our animal facility. We try to keep the male mice group housed 3-5 per cage whenever possible. Our largest problem is that the group housed male mice in our studies are most often fighting and needs to be split up to single cages, and to not get to large variations between single housed and group housed male mice in for example behaviour studies we need to single house also the male cages where there is not any fighting. This means that we end up with a lot of single housed males, which is not so ethical especially for longer term studies where we keep the mice up to 12 months. Since we don't have ethical approval to keep large cohorts of mice single housed for several months, we tried co-housing with a female mouse and to avoid uncontrolled breeding and extra pups we needed the females to be ovariectomized. We have added the rational for the experimental groups in the introduction to clarify why we have chosen to compare group housed male mice and male mice pair housed with ovariectomized females line 96-100: “In several of our studies we do not have ethical permission to single house male mice, at the same time as we are experiencing problems with fighting in group-housed male mice. Our alternatives for housing is therefore either to try to keep the male mice group housed (if they don’t hurt each other) or pair-house them with an ovariectomized female.

 In our breeding unit we get a lot of surplus female mice so there is no need for buying in extra mice, but of course they have to go through this minor surgery which takes about 10 minutes per mouse and a couple of days of recovery for the females. The females are then reused as companions in other studies throughout their life span to maximize the use of each female mouse. However, we agree that this is an ethical dilemma to use ovariectomized females as a companion to reduce the stress of single housed males and explain that in the discussion section line 431-435: However, there is of course an ethical dilemma in using extra female mice that undergo a small surgical intervention, to increase the animal welfare of the single housed laboratory male mice. By using surplus females from our breeding unit and use them as companions in several studies throughout their life span, we could maximize the benefit of each animal.

Reviewer 2 Report

Comments and Suggestions for Authors

The manuscript describes a study showing that when male mice are grouped with ovariectomized females, there is no aggression or fighting between them and that these male mice also show better memory and tend to be less stressed than animals that remain in groups.

The authors suggest that the use of ovariectomized females could be useful to increase the welfare of the animals (which I agree with in certain circumstances), but in the manuscript there is, for example, not enough discussion of the behavioral results, but mere speculation. It is not clear what it means that paired males have better memory than those in groups, and especially how this correlates with better welfare or new housing strategies.

On the other hand, I believe that the methods are not sufficiently well described. For example: the nomenclature of the mouse strain is incorrect; the procedure for ovariectomies is not described at all; at no point is the time the animals are grouped detailed; the physiological significance of the zero maze test is not explained; the blood collection technique is not sufficiently described, etc.

As for the results, I believe that results that are not sufficiently relevant (nor statistically significant) are overestimated, and vice versa.

In summary, I believe that this manuscript has many shortcomings and would need a very thorough revision in order to be published.

Author Response

The manuscript describes a study showing that when male mice are grouped with ovariectomized females, there is no aggression or fighting between them and that these male mice also show better memory and tend to be less stressed than animals that remain in groups.

The authors suggest that the use of ovariectomized females could be useful to increase the welfare of the animals (which I agree with in certain circumstances), but in the manuscript there is, for example, not enough discussion of the behavioral results, but mere speculation. It is not clear what it means that paired males have better memory than those in groups, and especially how this correlates with better welfare or new housing strategies.

Thank you for pointing out that more discussion is needed for the behavioural results.  We have now added more discussion about the link between stress and memory to clarify how that is connected to the housing conditions and added a new reference on this topic in line 403-407:Group housed male mice form a hierarchy within the group, which also affects the stress levels of the mice, depending on their ranking in the group. There is a clear link between social stress and memory (ref. Battiavelli D et al. Biological Psychiatry, 2024), which means that the social stress caused by the group housing condition can be the reason for the lower performance in the memory test compared to the males that were pair-housed with ovariectomized females.

On the other hand, I believe that the methods are not sufficiently well described. For example: the nomenclature of the mouse strain is incorrect; the procedure for ovariectomies is not described at all; at no point is the time the animals are grouped detailed; the physiological significance of the zero maze test is not explained; the blood collection technique is not sufficiently described, etc.

Thank you for your valuable feedback. We have made some improvements as following:

  • We have written the full nomenclature of the mouse strain “C57BL/6NCrl” throughout the manuscript: line 148, 150, 159 and 444.
  • We have added the procedure of ovariectomy in line 152-156 as following: “Ovariectomy was performed by making a dorsal skin incision in the female mice. A blunt puncture through the abdominal wall was performed on each side and the ovaries were removed through it. The fallopian tube was cauterized subsequently, and the skin incision was closed with wound clips.
  • We have now clarified that the male mice were already group housed from weaning age in line 149: “The male mice were already housed in groups of four from weaning age…
  • We also made some changes in zero maze description to clarify it’s physiological significance in line 209-221: “The zero maze test was performed to study anxiety related behavior in the mice (33, 34). The zero maze (Accuscan Instruments Inc., Columbus, Ohio, USA) is a circular black acrylic platform which is 5 cm wide, has an inner diameter of 40 cm and is elevated 75 cm from the floor. The maze is equally divided into four areas consisting of two oppo-site-sides which are covered by 30 cm acrylic transparent wall (closed arms) and two other arms that are open. The maze is equipped with photocell transceivers which monitors the mouse activity when it goes in and out from the closed arms. During the experiment, the mouse was placed on one of the closed arms and the test was performed during 5 minutes on two consecutive days. The activity in the closed and open arms, the time spent in both arms as well as the latency to enter the open arm for the first time were recorded. Mice who are spending more time in the closed arms are interpreted as more anxious than mice spending less time in the closed arms.
  • The blood collection technique and timing of the blood sampling is now better described in line 224-230: “Blood was collected for corticosterone measurement in the beginning of study (when mice are 6 weeks of age) and before termination of mice. Blood sampling was performed between 09.00 am to 11.00 am on both baseline and terminal sampling. Mice were placed in a restrainer (Agnthos AB, Lidingö, Sweden), the distal part of the tail vein was briefly cut using a sterile scalpel blade and 20 µl blood was collected in EDTA tubes (Satstedt, Nümbrecht, Germany). After the tail cut, the bleeding was stopped by applying a gentle pressure with a sterile pad.

As for the results, I believe that results that are not sufficiently relevant (nor statistically significant) are overestimated, and vice versa.

We have upon a suggestion from our statistical support added a calculation of the change in corticosterone levels between baseline and end of study, which shows that the decrease in corticosterone levels is significantly larger in the pair housed mice compared to the group housed mice, which gives us more confidence to discuss that the pair-housing condition is less stressful for the mice compared to group-housing. This new result can be found in Figure 5, described in line 341-344:The corticosterone ELISA assay results showed that there is significantly different between the change of corticosterone level in pair housed and group housed. The reduction in corticosterone levels is significantly larger in the pair-housed males than in the group housed males (Figure 5).  and discussed in line 418-421: “Lastly, from the hormone measurement, the change of corticosterone level differs significantly between two experimental groups. The pair housed group had significantly larger reduction of corticosterone level from baseline towards the end of the study. The finding indicates that the pair housed mice are less stressed than the group housed mice.” We have also rephrased this finding in Simple Summary line 18-19: “Regarding stress, pair housed male mice had a larger reduction in corticosterone levels during the study indicating lower stress level.”, in Abstract line 31-32 “Subsequently, the pair housed male mice had a larger reduction in corticosterone levels compared to group housed males.” and in line 373-374: “We found that the pair housed male mice demonstrated a better memory, showed larger decrase in corticosterone hormone levels in plasma from baseline to the end of the study and had a tendency of less anxiety-like behavior compared to the group housed male mice.”

We have also adjusted our phrasing of not statistically significant results to not overestimate the interpretation of these data in the discussion. For example we have exchanged the subheading from describing a tendency of increased time in open arms in pair housed mice to describing a significant result about the decreased activity of the pair housed mice in line 313: “The pair housed mice are less active in the zero maze.” and pointed out that there is only a tendency towards lower anxiety in line 421-423:Taken together it seems like male mice pair housed with ovariectomized females have lower stress levels and a tendency towards lower anxiety compared to group housed male mice.“

Reviewer 3 Report

Comments and Suggestions for Authors

Improved memory and lower stress levels in male mice co-housed with ovariectomiced female mice

In groups of male mice fighting takes often place, that causes injuries and stress. Research to avoid fighting by improving the management of housing conditions is therefore meritorious.

For further improvement of the manuscript, please add the time of blood collection as mice have a pronounced circadian rhythm of corticosterone concentration.

Author Response

Improved memory and lower stress levels in male mice co-housed with ovariectomiced female mice

In groups of male mice fighting takes often place, that causes injuries and stress. Research to avoid fighting by improving the management of housing conditions is therefore meritorious.

For further improvement of the manuscript, please add the time of blood collection as mice have a pronounced circadian rhythm of corticosterone concentration.

Thank you for pointing out the missing information. We have added the time of blood sampling in line 225-226: “Blood sampling was performed between 09.00 am to 11.00 am on both baseline and terminal sampling.

Reviewer 4 Report

Comments and Suggestions for Authors

It has been well acknowledged that housing conditions pose significant effect on mice, especially singly housing, which induces stress in mice and followed by anxious-depressive-like behavioral phenotypes. Moreover, once separate for a few days, male mice will fight aggressively with their previous cage mates if housed in the same cage again. Not being able to be group housed after singly housing has been one of the biggest challenges to use male mice to study effects of housing conditions and social behaviors. In the current manuscript, Wikanthi et al examined effects of housing condition on male mice when paired housed with another ovariectomized famale or group housed with four other male mice. Their findings about housing with an ovariectomized famle may reduce stress and anxiety seem interesting. However, there are significant flaws need to be addressed in the experimental design and over-interpretations in the conclusions need to be changed prior to be accepted for publication.  

1.     Appropriate control groups and control experiments are missing. Housing effect was compared between mice housed with 1 ovariectomized female and 4 regular male mice, which makes no sense. At lease pair-house with 1 regular male mouse is needed for the manuscript. Pair-house with 1 regular female mouse would add more credits to the results of the current study, if mating behavior can be prevented.

2.     None of the measures obtained in zero-maze and corticosterone level reached statistical significance, therefore, cannot be interpreted as different. The authors only performed t-test for all the statistical analyses, which should be replaced with tests correcting for multiple comparisons.  

3.     Results from passive avoidance test may be confounded by stress effect in the current setting, therefore, are not sufficient to conclude as changes in memory. Behavioral tests known for their validity in testing memory-related functions are required to provide convincing evidence.

Comments on the Quality of English Language

English needs to be polished to avoid misinterpretation and grammar errors

Author Response

It has been well acknowledged that housing conditions pose significant effect on mice, especially singly housing, which induces stress in mice and followed by anxious-depressive-like behavioral phenotypes. Moreover, once separate for a few days, male mice will fight aggressively with their previous cage mates if housed in the same cage again. Not being able to be group housed after singly housing has been one of the biggest challenges to use male mice to study effects of housing conditions and social behaviors. In the current manuscript, Wikanthi et al examined effects of housing condition on male mice when paired housed with another ovariectomized famale or group housed with four other male mice. Their findings about housing with an ovariectomized female may reduce stress and anxiety seem interesting. However, there are significant flaws need to be addressed in the experimental design and over-interpretations in the conclusions need to be changed prior to be accepted for publication. 

  1. Appropriate control groups and control experiments are missing. Housing effect was compared between mice housed with 1 ovariectomized female and 4 regular male mice, which makes no sense. At lease pair-house with 1 regular male mouse is needed for the manuscript. Pair-house with 1 regular female mouse would add more credits to the results of the current study, if mating behavior can be prevented.

We understand the point of comparing the male mouse that is pair housed with an ovariectomized female with a male that is only pair housed, and not group housed, with one male. The background to performing the study described in our manuscript, was to compare our regular housing condition, having 4 male mice housed together, with a housing condition where we house a male with an ovariectomized female. The idea was that when the male mice are housed together with other males and a fight is started, the males would not need to be separated and housed alone in a cage without any companion, rather be housed with a female with no risk of fighting nor generating pups in an uncontrolled way. We chose the best pair housing partner from our own experience and published paper, where different pair housing companions were investigated (Ewaldsson et al, Scandinavian Journal of Laboratory Animal Science, 2016, Volume 42, Number 7, Who is a compatible partner for a male mouse?). We have added the rational for the experimental groups in the introduction to clarify why we have chosen to compare group housed male mice and male mice pair housed with ovariectomized females line 96-100: “In several of our studies we do not have ethical permission to single house male mice, at the same time as we are experiencing problems with fighting in group-housed male mice. Our alternatives for housing is therefore either to try to keep the male mice group housed (if they don’t hurt each other) or pair-house them with an ovariectomized female.”

  1. None of the measures obtained in zero-maze and corticosterone level reached statistical significance, therefore, cannot be interpreted as different. The authors only performed t-test for all the statistical analyses, which should be replaced with tests correcting for multiple comparisons. 

We have upon a suggestion from our statistical support added a calculation of the change in corticosterone levels between baseline and end of study, which shows that the decrease in corticosterone levels is significantly larger in the pair housed mice compared to the group housed mice, which gives us more confidence to discuss that the pair-housing condition is less stressful for the mice compared to group-housing. This new result can be found in Figure 5, described in line 341-344:The corticosterone ELISA assay results showed that there is significantly different between the change of corticosterone level in pair housed and group housed. The reduction in corticosterone levels is significantly larger in the pair-housed males than in the group housed males (Figure 5).  and discussed in line 418-421: “Lastly, from the hormone measurement, the change of corticosterone level differs significantly between two experimental groups. The pair housed group had significantly larger reduction of corticosterone level from baseline towards the end of the study. The finding indicates that the pair housed mice are less stressed than the group housed mice.” We have also rephrased this finding in Simple Summary line 18-19: “Regarding stress, pair housed male mice had a larger reduction in corticosterone levels during the study indicating lower stress level.”, in Abstract line 31-32 “Subsequently, the pair housed male mice had a larger reduction in corticosterone levels compared to group housed males.” and in line 373-374: “We found that the pair housed male mice demonstrated a better memory, showed larger decrease in corticosterone hormone levels in plasma from baseline to the end of the study and had a tendency of less anxiety-like behavior compared to the group housed male mice.”

For the zero maze, we admit that it was not described in detail why we did the calculations as we did. Even if the zero maze test was performed on two consecutive test days (day 1 and 2), each test day is a separate test. The first day, the test evaluates the performance in a novel environment and the second day, the test evaluates the performance in a familiar environment. Therefore, we performed a t-test for each parameter separately for the two days and did not combine or calculate statistics on the data from both days together. Regarding the measures in the zero maze, the lower activity in pair-housed mice on day 1, and the increased latency to enter the open arms of the pair housed mice on day are both significant (p>0.01). We agree that the time spent in the open and closed arms in the zero maze are not significantly different between the groups (although the p-values are close to significant; 0.054 and 0.056) and we have adjusted our phrasing of not statistically significant results to not overestimate the interpretation of these data in the discussion. For example we have exchanged the subheading from describing a tendency of increased time in open arms in pair housed mice to describing a significant result about the decreased activity of the pair housed mice in line 313: “The pair housed mice are less active in the zero maze.” and pointed out that there is only a tendency towards lower anxiety in line 421-423:Taken together it seems like male mice pair housed with ovariectomized females have lower stress levels and a tendency towards lower anxiety compared to group housed male mice.“

  1. Results from passive avoidance test may be confounded by stress effect in the current setting, therefore, are not sufficient to conclude as changes in memory. Behavioral tests known for their validity in testing memory-related functions are required to provide convincing evidence.

Thank you for pointing out the link between stress and performance in the memory test. We agree that the performance in the memory test is likely affected by the stress levels of the mice, which then can be connected to the housing conditions. From our perspective this is of interest that stress levels can affect the outcome of this fear-conditioned memory test, but we don’t claim that it’s affecting other types of memories that we are not testing here like spatial memory etc. We have now clarified the link between stress and outcome of the memory test in the discussion in line 403-408: “Group housed male mice form a hierarchy within the group, which also affects the stress levels of the mice, depending on their ranking in the group. There is a clear link between social stress and memory (ref Battivelli D et al. Biological Psychiatry, 2024), which means that the social stress caused by the group housing condition can be the reason for the lower performance in the memory test in our study compared to the less stressed males that were pair-housed with ovariectomized females.

Round 2

Reviewer 2 Report

Comments and Suggestions for Authors

The authors have improved the manuscript substantially in this second version, clarifying many points that were not sufficiently developed in the first one. I think it is now much clearer what the authors want to explain and discuss.

However, as far as the description of the ovariectomy technique is concerned, I still miss important details such as anesthesia and analgesia regimes, recovery time, postoperative care, etc.

Apart from these specific points, which I think it is important and necessary to include, I consider this version of the manuscript suitable for publication.

Author Response

The authors have improved the manuscript substantially in this second version, clarifying many points that were not sufficiently developed in the first one. I think it is now much clearer what the authors want to explain and discuss.

However, as far as the description of the ovariectomy technique is concerned, I still miss important details such as anesthesia and analgesia regimes, recovery time, postoperative care, etc.

Thank you for pointing out the important details that were still missing in the description of the ovariectomy surgery. It is now described in more detail in page 3 and 4, line 127-136.

Ovariectomy surgery was done by Charles River (L’Arbresele, France). Before surgery started, the surgical area was shaved and disinfected with a aseptic solution. Eye gel was applied to protect the eyes of the mice. Analgesia (Buprenorphine at 0.1 mg/kg) was injected subcutaneously before surgery. The mice were anaesthetized with isoflurane. Ovariectomy surgery was performed by making a dorsal skin incision in the skin. A blunt puncture through the abdominal wall was performed on each side and the ovaries were removed through it. The fallopian tube was cauterized subsequently, and the skin incision was closed with wound clips. After surgery, clinical examination was performed at the same occasion as injection of analgesia. Buprenorphine was injected 4 hours after surgery. In addition, NSAIDs (Carprofen at 5 mg/kg) was injected subcutaneously 24 hours and 48 hours after surgery. Heating pad was utilized during whole recovery time. The ovariectomized female mice were recovered after 3 to 5 days.

Reviewer 4 Report

Comments and Suggestions for Authors

Interpreting the results from zero-maze for both anxiety-related and memory-related contexts seem to go too far from the actual data. Please provide references to support the interpretation of day2 results (especially latency to enter the open arm) as a measurement for memory related functions. Otherwise, please focus on anxiety-related functions only. Additionally, increased latency to the open arm and increased time spent on the open arm were both present in day 2 from pair-house group, which is already contradicting each other in terms of anxiety. Please elaborate on this data in your discussion as well.

I have noticed that fractions are not correctly presented in the manuscript, for example 0,054 should be presented as 0.054. Please correct them.

Comments on the Quality of English Language

English needs significant improvement, especially grammar and structure of the sentences. For example, 'showed that there is significantly different 313 between the change of corticosterone level in pair housed and group housed', either there is significantly difference or they are significantly different. 

Author Response

Interpreting the results from zero-maze for both anxiety-related and memory-related contexts seem to go too far from the actual data. Please provide references to support the interpretation of day2 results (especially latency to enter the open arm) as a measurement for memory related functions. Otherwise, please focus on anxiety-related functions only.

Thank you for making us aware but we do not have any reference to support the memory-related interpretation of data in the discussion. That was a speculation from our side and have now been removed from the manuscript on page 9, lines 380-386 (also found below) and we are focusing on the anxiety-related functions only.

The pair housed mice also displayed an increased latency to enter the open arm in the zero maze on the second day compared to the group housed mice (Figure 4b). The finding indicates that the pair housed mice remembered the platform from day 1 and were less interested in exploring the platform again in day 2. In summary these findings indicate a better memory in the male mice pair housed with ovariectomized females compared to group housed male mice.

In addition and and to clarify, we have added a description the difference between the first and second day measurements in the Zero Maze in page 5, line 199-200 in the manuscript.

During the experiment, the mouse was placed on one of the closed arms and the test was performed during 5 minutes on two consecutive days, first day for measurements in a novel environment and the second day in a familiar environment.

Additionally, increased latency to the open arm and increased time spent on the open arm were both present in day 2 from pair-house group, which is already contradicting each other in terms of anxiety. Please elaborate on this data in your discussion as well.

Thanks for pointing out this contradiction. We have removed below sentences that was previously found in page 8 line 380-386 in the manuscript.

The pair housed mice also displayed an increased latency to enter the open arm in the zero maze on the second day compared to the group housed mice (Figure 4b). The finding indicates that the pair housed mice remembered the platform from day 1 and were less interested in exploring the platform again in day 2. In summary these findings indicate a better memory in the male mice pair housed with ovariectomized females compared to group housed male mice.

We have also added the below text to page 8-9, lines 379-382 in the manuscript.

In addition, a longer latency to enter the open arm the second day, in a familiar environment, could be an indication of lower stress levels rather than indicating that the pair housed mice would be more anxious since the environment is familiar and there is no need to explore it to the same extent as in the first day when the environment is novel.

I have noticed that fractions are not correctly presented in the manuscript, for example 0,054 should be presented as 0.054. Please correct them.Why todays, describe in manuscript

Thank you for noticing the wrongly written commas, they are now changed to dots instead in the graph in page 7, line 299.

English needs significant improvement, especially grammar and structure of the sentences. For example, 'showed that there is significantly different 313 between the change of corticosterone level in pair housed and group housed', either there is significantly difference or they are significantly different. 

Thanks for pointing out that the English in our manuscript needs to be improved. The manuscript has now been revised by a colleauge fluent in English writing.